



**China Wildfire Emission (ChinaWED v1) for the period 2012-2022**
Zhengyang Lin[a], Ling Huang[a], Hanqin Tian[b], Anping Chen[c], Xuhui Wang[a*]
a.   Institute of Carbon Neutrality, Sino-French Institute of Earth System Sciences,
College of Urban and Environmental Sciences, Peking University, Beijing, China
b.   Center for Earth System Science and Global Sustainability, Schiller Institute for
Integrated Science and Society, Department of Earth and Environmental Sciences,
Boston College, Chestnut Hill, MA, USA
c.   Department of Biology and Graduate Degree Program in Ecology, Colorado State
University, Fort Collins, USA
*Correspondence to*: Xuhui Wang(xuhui.wang@pku.edu.cn)
## Abstract
During the past decades, wildfires have undergone rapid changes while both the
extent of fire activities and the resulting greenhouse gas (GHG) emissions from
wildfires in China remain inadequately quantified. To explore national wildfire-induced
emissions, we employed satellite-based data on burned vegetation to generate the
China Wildfire Emission Dataset (ChinaWED). This dataset is constructed at monthly
and kilometer scale under a consistent and quantifiable calculation framework,
providing an average annual estimates of wildfire-induced GHG emissions of 78.13 ±
22.46 Tg $CO_2$, 279.47 ± 82.01 Gg $CH_4$, and 6.26 ± 1.67 Gg $N_2O$ for the past decade.
We observed significant decreases in both wildfire occurrences and emissions within
forests and grasslands. This trend, however, is counteracted by increasing agricultural
fires, which constitute the primary type accounting for at least half of the national total
fire emissions. The seasonal cycle of wildfire GHG emissions show an evident apex
occurring during the transition from mid-spring to early-summer. At the regional scale,
Northeast, Southwest and East China emerge as hotspots for wildfire-induced
emissions. Our study offers new insights into understanding China's wildfire dynamics
and provides a detailed regional model for the wildfire greenhouse gas emissions over
China.



## 1. Introduction


Wildfires exert a substantial impact on landscape vegetation while influencing the
biogeochemical cycle through the emissions of greenhouse gases (GHG) (Bauters et
al., 2021; Guo et al., 2024; Rodríguez Vásquez et al., 2021). Approximately $2.1 \times 10^{15}$
grams (Petagrams, Pg) of carbon were emitted globally through biomass burning,
representing about 22% of all fossil fuel emissions in 2021 (Friedlingstein et al., 2022;
van Wees et al., 2022; van der Werf et al., 2017). It constitutes a crucial component of
the global and regional GHG budget (carbon dioxide ($CO_2$), methane ($CH_4$) and nitrous
oxide ($N_2O$)), which is of particular concern giving 120 countries have pledged to
achieve net zero GHG emissions. China, in particular, announced and initiated long-
term climate plans, aiming for carbon peaking by 2030 and carbon neutrality by 2060
(Liu et al., 2022). Additionally, over the past decade in China, climate-driven fire
weather, expanding vegetation-based fuel loadings, and anthropogenic activities have
led to rapidly changing fire dynamics (Wang et al., 2023a; Wiedinmyer et al., 2023;
Ying et al., 2018). To address the challenge and achieve the goals, one key step is to
establish a national scale dataset that reflects the recent wildfire emission dynamics
and contributes to the domestic GHG budget (Friedlingstein et al., 2022).
Currently, there have been different studies working on the estimates of China
wildfire emissions including contributions from some global products. One of the most
widely-used approaches take the product of emission factors, fuel loadings, burned
area and combustion efficiency as the estimate of emissions. It should be noted that
the limitations stem from various aspects during the calculation steps. For example,
these studies may use the universal parameters (e.g., land cover types, emission
factors) that do not match with characteristics of local fuels and further estimates (van
Wees et al., 2022; Wiedinmyer et al., 2023). Uncertainty also arises from estimates of
burned area due to the remote sensing-based fire datasets with different emphasis
(e.g., active fire product and burned area product) (Chen et al., 2020; Giglio et al., 2018;
Schroeder et al., 2014). Some other research focused on agricultural fire emissions
adopted traditional "crop-yield-based approaches" (CYBAs), primarily relying on
provincial statistical data and field-reported measurements such as crop production
and estimates of burned crop residues (Hong et al., 2023; Li et al., 2016). These parts
are hard to verify and can only be measured within administrative boundaries. In
addition, the estimates from CYBAs typically have relatively long updating cycles, often
on a yearly scale. These approaches form the fundamental framework of emission
estimates, yet various input parameters were incorporated and the emissions of GHGs
may not be consistent even within products.
Here, we present the China Wildfire Emission Dataset (ChinaWED v1) for the
period from 2012 to 2022 at monthly and kilometer scale. We focused on the limitations
existing in current studies and products and refined the estimates of calculation
components. Emission factors that are specifically suited for evaluating wildfire
emission in China retrieved from previous studies conducted domestically and in
neighboring countries were collected. Previous studies have reported a majority of
wildfire occurrences in croplands, highlighting the need for improved burned area



estimates that incorporate small-size fire activities (Ying et al., 2021; Zhang et al.,
2015). The newly developed product is easily to update with only one-month to two-
months lag and provide consistent results for all three GHGs under same calculation
framework. With the support of this ChinaWED product, we can also capture and
explore the magnitude, patterns, trends and drivers of the wildfire occurrences and the
wildfire-induced emissions in China within the past decade.

## 2. Methods

### 2.1 Emission estimation

In this study, we adopted the wildfire emissions estimation method based on the
combination of four components: burned area, fuel load, emission factor and
combustion completeness, calculated by the following equation:

$$E_{i,x,t} = \sum_{j}^{n} BA_{t,x} \times FL_x \times EF_{i,j} \times CC_{x,j} \tag{1}$$

where the subscript $i$ represents specific emission types, $j$ represents different
vegetated cover types, $x$ and $t$ stand for spatial and temporal information; $E_{i,x,t}$ is
hence the estimated amount of emission type $i$ in location $x$ and month $t$; $BA_{t,x}$ is
the total aggregated burned area derived from multisource of satellite-based products
in location $x$ and month $t$; $FL_x$ is fuel load in location $x$; $EF_{i,j}$ is emission factor of
specific emission type $i$ for vegetated cover type $j$; $CC_{x,j}$ is defined as combustion
completeness in location $x$ for vegetated cover type $j$.

### 2.2 Burned area calculation

Satellite-based thermal anomalies include burned area and active fire products,
equipping researchers with the capability to observe these distinctive signatures
across extensive spatial and temporal ranges. Burned areas are determined by
analyzing the disparities in visible and near-infrared channels between pre- and post-
fire satellite images. One of the most common limitations in burned area products is
the exclusion of small-sized or smoldering fires. In contrast, active fire detection is
capable of sensing these fires benefitting from the use of the thermal-sensitive mid-
infrared channel. Here we use MODIS burned area product and achieved FIRMS
VIIRS S-NPP active fire records   as the main input datasets (Giglio et al., 2018;
Schroeder et al., 2014).
MCD64A1 provides burned area classification at 500 m spatial resolution and
monthly temporal resolution. VIIRS S-NPP provides daily active fire detection at 375
m spatial resolution. Given active fire detection's capability to identify fires occupying
5% or less of a pixel, the S-NPP active fire records can provide more detailed
information, particularly in regions like China where numerous crop residue burnings
occur. Current models and studies counted the active fire points located outside
existed burned area directly as the supplementary sources for the fire activities. To
avoid the potential excessive measurement, a reanalysis system combining both





burned area and active fire was designed and demonstrated in Fig S1. We
reconstructed the external burned area derived with circular kernels centered at those
active fire records. The aggregated burned area is calculated as below:
$$BA_{t,x} = BA_{main(t,x)} + \sum_{m}^{n} AF_{sf(t,x,m)} \qquad (2)$$
where the subscript and left part of the equation is same with that in equation (1);
$BA_{main(t,x)}$ represents the burned area cells in location $x$ and month $t$; the sum of
$AF_{sf(t,x,m)}$ represents potential burned area determined through the counting of
decomposed small pixels from circular kernels centered at those active fire records
(Fig. S3 and Fig. S4).
Additionally, we incorporated an independent inventory of fixed-location heat
sources. This inventory is featured by continuously operating heat-source objects and
spatiotemporal-aggregation characteristics in thermal anomalies. It encompasses
heat-source objects including active volcanos, industrial heat sources (e.g., coal-
related plants, nonmental mineral producing, ferrous metal related plants) (Liu et al.,
2018). We utilized this inventory as a filter to exclude "burned areas" pixels that are
not caused by wildfires. Finally, the processed burned area results were resampled to
1 km spatial resolution to match the fuel load and land cover mapping. In general,
nearly three quarters (76.2%) of the total burned area is derived directly from the
MCD64 burned area product, while 24.5% is supplemented by information from VIIRS
S-NPP 375 m active fire records Through the incorporation of an independent fixed
heat source dataset, we were able to filter out 0.7% of the burned area.

**2.3 Calculation of other components**
Prior studies integrated upscaled systematic field investigations and regional or
national censuses to map the fuel load. Recent results showed that AGB can serve as
a proxy observation, enabling indirect estimations of dry matter. Remotely sensed
biomass carbon density maps aiming at limited vegetation types have been widely
used. Here we used the newly developed 300 m spatial resolution dataset from Spawn
et al. that incorporates multisource previously presented biomass map and harmonizes
AGB from different vegetation types (Noon et al., 2022; Spawn et al., 2020).
We used land cover product from the ESA Climate Change Initiative to describe
the different vegetation types (Li et al., 2018). This product has identical spatial
resolution to this harmonized AGB dataset. We further aggerated the initial 37 classes
into three major vegetated categories, namely forests, herbaceous and cropland. To
refine the estimation of crop residue burning, several independent datasets of high-
resolution crop type mapping are utilized as well. These dataset contain spatial
distribution of double season paddy rice (Pan et al., 2021), single season rice (Shen
et al., 2023), maize (Shen et al., 2022), winter wheat (Dong et al., 2020) and sugarcane
(Zheng et al., 2022) with 10 m or 20 m spatial resolution.
It should be noted that the resolution of all these above datasets were downscaled
to 1 km. AGB was calculated by summing all pixels, land cover was determined based
on the mode value of vegetated categories, and detailed crop types were identified by
counting classified pixels. AGB provided consistent and seamless estimations of
biomass carbon density globally for the fixed year 2010. Land cover data were
computed from 2001 to 2020, while crop type mapping was primarily calculated
between 2017 and 2020. We utilize annual land cover data associated with the burned
area for the corresponding year (mapping the burned area in 2020 for the period from
2020 to 2022). For distinct crop types, we specifically employ the results obtained
during their respective growing seasons, coupled with the monthly burned area data.
The averaged multiyear crop type mapping was harmonized into land cover data where
agricultural land use pixels were present.
Different previous studies applied constant thresholds which is considered a major
bias in emission estimation (Zhang et al., 2008). We adopted a method based on the
combination of land cover types and fraction of burned (FB) assigned as a function of
tree cover (Wiedinmyer et al., 2023; Wu et al., 2018; Zhang et al., 2011). Agricultural
land use was set to fixed combustion completeness value to 0.93. Herbaceous had
similar high CC values defined by the fraction of tree coverage while forests had much
lower CC values. The detail values are listed in Table S.1 .
Emission factors for different vegetation and emission types were summarized in
Table S.2. Apart from the studies that introducing global fire emissions, we selected
publications that focused on affected burned areas in China and neighboring countries.
Detailed emission factors of different crop types were one of the primary objectives
and used in this study to help improve our burned area-based emission estimation.
Forests were divided into tropical, temperate and boreal types, identified by the
updated digital Köppen–Geiger world map of climatic classification (Beck et al., 2018).

## 3. Results

### 3.1 Characteristics of China wildfires and emissions

ChinaWED was calculated based on a burned area-based approach. We
integrated different remotely sensed datasets that map regions affected by wildfires
and detect active fire spots to reconstruct the burned area. From 2012 to 2022, the
total burned area in China amounted to 5.31 ± 1.70 million hectares per year (Mha yr$^{-1}$)
(Fig. 1). More than four-fifths of the total burned area were located in croplands,
equivalent to the land area of Switzerland. 11.0% of the burned area occurred in
various types of forests, while less than 6% of the burned area took place in grasslands
or other herbaceous-dominated regions. Based on this burned area estimates and
calculation of other components (emission factors, fuel loads, etc. see methods), our
results showed that annual wildfire-induced GHG emissions in China amounted to
78.13 ± 22.46 Tg $CO_2$, 279.47 ± 82.01 Gg $CH_4$, and 6.26 ± 1.67 Gigagrams (Gg) $N_2O$
(Fig. 1). Although the majority of all wildfire-induced GHG emissions were still caused
by cropland fires, the proportions were quite different from that in burned area. A fifth
of $CO_2$ (21.1%) and $CH_4$ (19.9%) emissions were caused by forest fires, which was
almost double than the contribution of this type measured in area. This comes from the
differences in background fuel loads as measured in carbon pools between forests and
cropland, reported by research on China's terrestrial ecosystems (Tang et al., 2018).





An even more substantial proportion of national $N_2O$ emissions came from forest fires,
reaching up to 37.1% of the total(Fig. 1). Wildfire-induced $N_2O$ emissions are highly
dependent on the ratio of carbon to nitrogen in vegetation fuels, which was higher in
woody areas (Vernooij et al., 2021). In comparisons to wildfires on other land cover
types, grassland fires played a comparatively minor role in wildfire dynamics and
emissions.

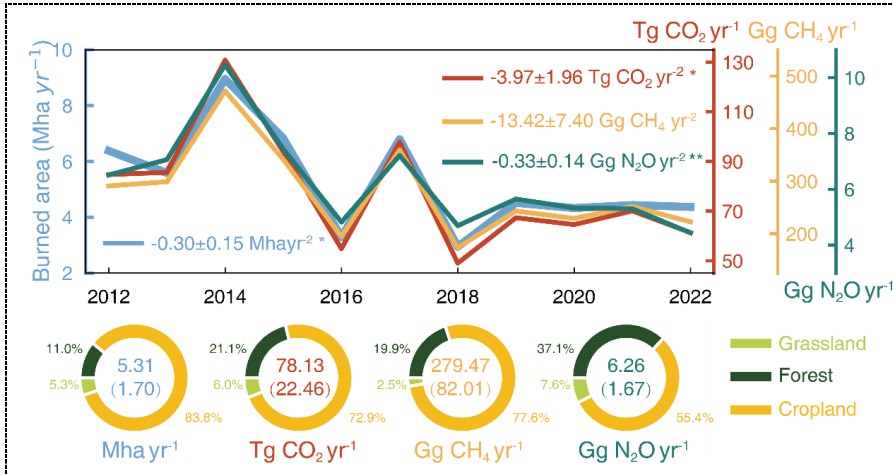

Fig. 1. The time-series and trends of China burned area and wildfire-induced emissions ($CO_2$, $CH_4$, $N_2O$) where significant trends are denoted by asterisks (*P < 0.1 and **P < 0.05). The bottom pie charts demonstrate the annual averages (standard deviation within the brackets) and the proportions of different land cover types during the study period.

During this period, the dataset recorded a decline trend of $-0.31 \pm 0.15$ Mha $yr^{-2}$
($P<0.1$) (Fig. 1). All vegetation wildfires decreased at different magnitudes, resulting in
pervasive and slightly different declines in the three greenhouse gases. Agricultural
fires had been gradually limited and demonstrated a decline in burned area at $-0.26 \pm$
$0.14$ Mha $yr^{-2}$. Affected by the variations in cropland burned area, the three types of
GHGs in our dataset had a non-significant decline at $-2.41 \pm 1.81$ Tg $CO_2$ $yr^{-2}$, $-8.97 \pm$
$6.96$ Gg $CH_4$ $yr^{-2}$ and $-0.15 \pm 0.11$ Gg $N_2O$ $yr^{-2}$ during the study period. Compared with
cropland, burned area and all three types of wildfire-induced greenhouse gases in
forests and grasslands dropped significantly and rapidly. The decline in forest fires
contributed to nearly a third ($CO_2$ at $-1.22 \pm 0.36$ Tg $yr^{-2}$, $P<0.01$ and $CH_4$ at $-3.93 \pm$
$1.21$ Gg $yr^{-2}$, $P<0.05$) and a half ($N_2O$ at $-0.15 \pm 0.05$ Gg $yr^{-2}$, $P<0.05$) in the total trends
of emissions (Fig. S1). The grassland contributed smaller in all these GHGs ($CO_2$ at -
$0.34 \pm 0.08$ Tg $yr^{-2}$, $P<0.01$, $CH_4$ at $-0.51 \pm 0.13$ Gg $yr^{-2}$, $P<0.01$, and $N_2O$ at $-0.03 \pm$
$0.01$ Gg $yr^{-2}$, $P<0.01$) within the past decade.



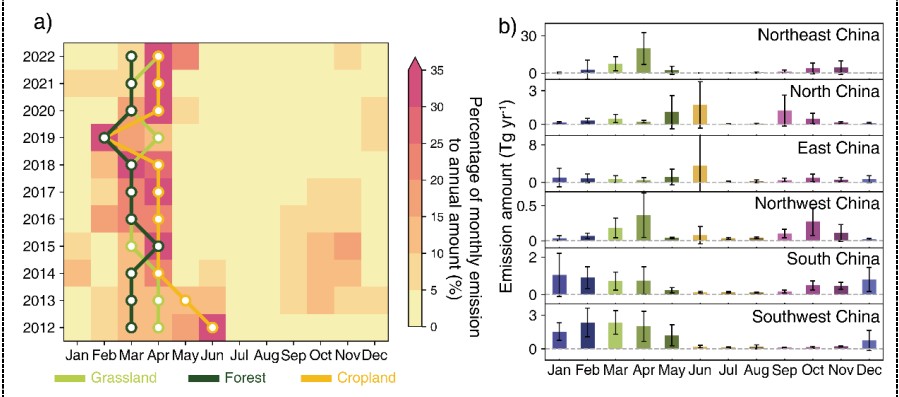

Fig. 2. Seasonal cycle of national and regional wildfire-induced emissions, taking $CO_2$ as an example. a) demonstrates the percentage of monthly emissions to annual amount. Vertical lines illustrate the peak emissions on different land cover types. b) demonstrated the average monthly $CO_2$ emissions in six regions, which are introduced in Fig. S2 and further described in Fig. 3. Emissions from six regions are depicted on distinct Y-axes to accurately capture the seasonal variations in emissions patterns. Four sets of colors indicate four seasons. The monthly calculation and seasonal/interannual patterns for wildfire-induced $CH_4$ and $N_2O$ emissions are demonstrated in Fig. S4 and Fig. S5.

The outcomes derived from diverse regions and land cover types underscored
those fires originating within cropland significantly dominated the overarching
dynamics of national wildfires and emissions. A spatiotemporal association was
assumed to exist between agricultural activities, particularly those related to planting
and harvesting preparations, and the incidence of wildfires. Throughout our study
period, the majority of all three types of GHG were concentrated in the first half of the
year. More than half of the annual $CO_2$ emissions from wildfires were observed from
late winter to middle spring (February to April), along with nearly the identical relative
proportions of $CH_4$ and $N_2O$. A secondary seasonal peak of wildfire-induced emissions
occurred in the harvest seasons in autumn (September to November), accounting for
nearly 20% of the annual total (Fig. 2a). We divided six specific wildfire-induced
emissions regions dependent on geographical location and environmental
characteristics (Fig. S2 and Table S3). The patterns of double peaks in agricultural fire
emissions in Northeast China had a significant impact on national emission levels.
During the major emission season, three quarters of the region's total annual amount
was emitted. It is important to note that the temporal patterns are closely associated
with the local sowing and harvesting seasons (Fig. 2b) (Cheng et al., 2022; WANG et
al., 2020). Similarly in North China, the major peak occurred in early summer (May and
June) while the secondary peak in mid-autumn (September and October). A total of
2.75 Tg and 1.65 Tg of annual $CO_2$ emissions induced by agricultural fires were
concentrated during these respective time periods. East China displayed disparate
seasonal patterns, with the majority of agricultural fires occurring during the summer
when the planting and harvest were made in double-season paddy rice fields in this
area (Fig. 2b) (Pan et al., 2021; Wu et al., 2023). Approximately one-third of the annual





regional emissions induced by wildfires were concentrated in June. Consequently, this
correlation is validated through the examination of seasonal cycles in wildfire
occurrences, which becomes a prominent temporal feature that drive the dynamics of
national-scale wildfire-induced emissions (Zhang et al., 2015).

**3.2 Spatiotemporal pattern of wildfire and its GHG emissions**
To further explore the fire emission dynamics, we calculated the provincial and
monthly burned areas and emissions, which were then aggregated to obtain regional
and seasonal statistics. The results showed that the national wildfire-induced
emissions shared similar patterns of all three GHG types in spite of their large
disparities at both spatial and temporal scales. More than four-fifths of the total of
domestic wildfire-induced GHG emissions (82.8% for $CO_2$, 83.2% for $CH_4$, and 83.6%
$N_2O$) located in three primary peaks, the Northeast, Southwest and East China,
respectively (Fig. 3), which will be introduced in detail in the upcoming sections.
In all six regions, Northeast China (Heilongjiang, Jilin, Liaoning and Nei Mongol)
affected by the highest wildfire emissions. Heilongjiang and Jilin were the top two
provinces not only within the region but also nationwide. Many of the burned area and
emissions located in vast plains (SongNen, Liaohe and Sanjiang plain) of Northeast
China. The vegetation-sourced fire emissions from these two provinces contributed to
nearly one-third and one-tenth of the total domestic emissions, individually. Moreover,
they exhibited a mild increasing trend compared to the national pattern, registering at
non-significant trends of $0.14 \pm 0.15$ Mha $yr^{-2}$ for burned area and $1.92 \pm 1.92$ Tg $yr^{-2}$,
$6.94 \pm 7.34$ Gg $yr^{-2}$ and $0.11 \pm 0.13$ Gg $yr^{-2}$ for $CO_2$, $CH_4$ and $N_2O$, respectively (Fig. 4).
According to results from the National Bureau of Statistics, the total of these four
provinces contributed a quarter in sown area and grain production for the past decade.
The extensive grain acreage and prevailing practices of burning crop straws for land
clearance jointly contributed to the high wildfire-induced emissions in agricultural land
uses in Northeast China. $CO_2$ emissions from crop residue burning accounted for 82.7%
of the regional total wildfire-induced emissions and 62.5% of the domestic emissions
for this type. The rising trends of agricultural fires constitute the majority of regional
wildfire dynamics.
Fires have been controlled to an average of 0.27 Mha of burned area per year
through systematic fire and forest management in this area (Fig. 3 and Fig. 4). For
comparison, a single fire event, namely the 1987 Great Black Dragon Fire, destroyed
1.33 Mha of forests and resulted in nearly two hundred fatalities (Zhao et al., 2020;
Zong et al., 2022). The boreal forest wildfires led to 5.28 Tg $CO_2$, 19.44 Gg $CH_4$ and
0.94 Gg $N_2O$, constituting 12.3% of the total wildfire-induced emissions of this region.
This amount was also equivalent for nearly ninety percent of the boreal forest wildfire
emissions nationwide. Grassland fires in Northeast China, specifically in the Hulun Buir
and Xilingol grasslands, attracted national attention, accounting significantly for the
total amount at 67.2% for burned area and 46.7% for wildfire-induced emissions
respectively.



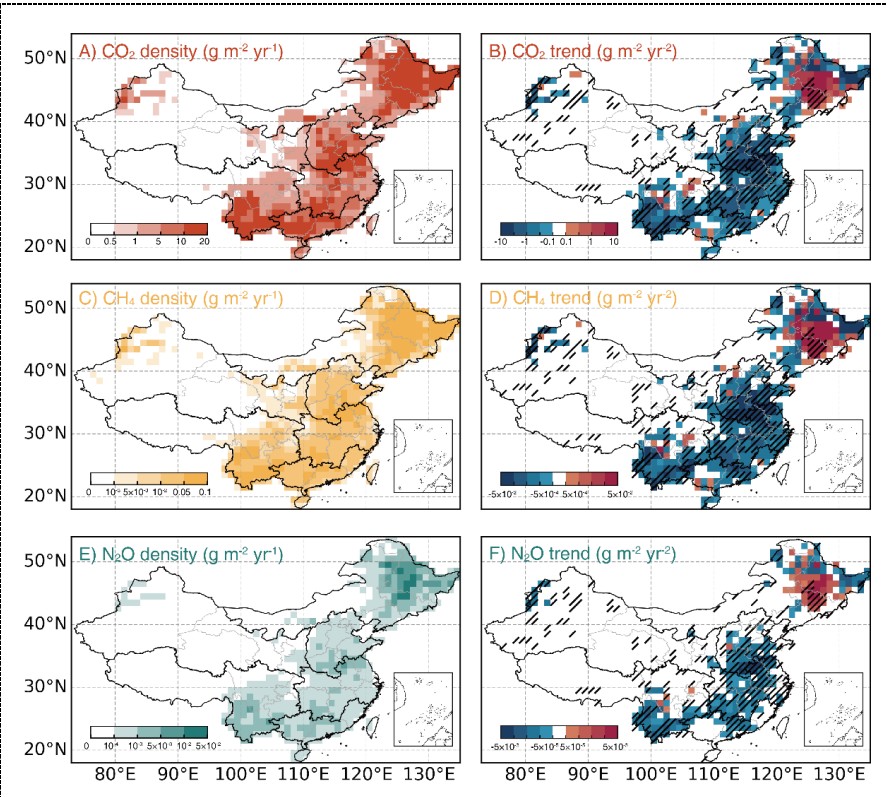

Fig. 3. Spatial distributions of the density (a, c, e) and the trends (b, d, f) of wildfire-induced emissions ($CO_2$, $CH_4$ and $N_2O$) with the same color showed in Fig. 1. To achieve better visual performance, the map demonstrated the density and trends in 1° grids where hatched area indicates significant trends ($P<0.05$). Locations of the provinces and regions are described in Fig. S2.

Southwest China, covering five provincial administrative areas (Yunnan, Sichuan
and Guizhou provinces, Chongqing and Xizang Autonomous Region), was the second-
largest regional scale emitter of fire-sourced greenhouse gases (Fig. 3). This region
stands out as the only area where agricultural wildfires do not dominate; instead,
temperate forest fires emitted more than all the other vegetation fires in this region (Fig.
4). Yunnan province, a pivotal player in shaping the wildfire dynamics of this region,
contributed substantially, with an annual burned area of 0.16 Mha, emitting 7.57 Tg
$CO_2$, 23.13 Gg $CH_4$, and 0.81 Gg $N_2O$ (Cui et al., 2022; Ying et al., 2021). These figures
accounted for over 60% of the regional burned area and wildfire-induced emissions.
From the perspective of recent trends, this province contributed to 82.4% of the
regional decrease in burned area and even larger share in the reduction of wildfire-
induced emissions. The border fires showed some shared similarities in fire spreading
mechanisms and environmental factors between this region and the adjoining Indo-
China Peninsula, a global wildfire hotspot. However, in comparison to the rapid land
cover changes and massive relevant wildfires reported in Southeast Asian countries,



involving activities such as slash-and-burn, commercial forest loss, and drainage in
peatlands (Curtis et al., 2018; Page et al., 2022), Southwest China had fewer and
weaker fire activities related with this type. The occurrences of forest fires usually arose
from occasional personal activities or fire-related cultural traditions (Ying et al., 2021).
On the other hand, due to recent implementations of fire policies and long-standing
efforts from firefighting teams, Southwest China has experienced a significant decline
in forest fires, with a decrease of -0.02 ± 0.00 Mha $yr^{-2}$ ($P$<0.01) for burned area and -
0.74 ± 0.23 Tg $yr^{-2}$ ($P$<0.05), -2.38 ± 0.74 Gg $yr^{-2}$ ($P$<0.05) and 0.09 ± 0.02 Gg $yr^{-2}$
($P$<0.05) for $CO_2$, $CH_4$ and $N_2O$, respectively. This reduction accounts for more than
65% of national declines in forest fires.

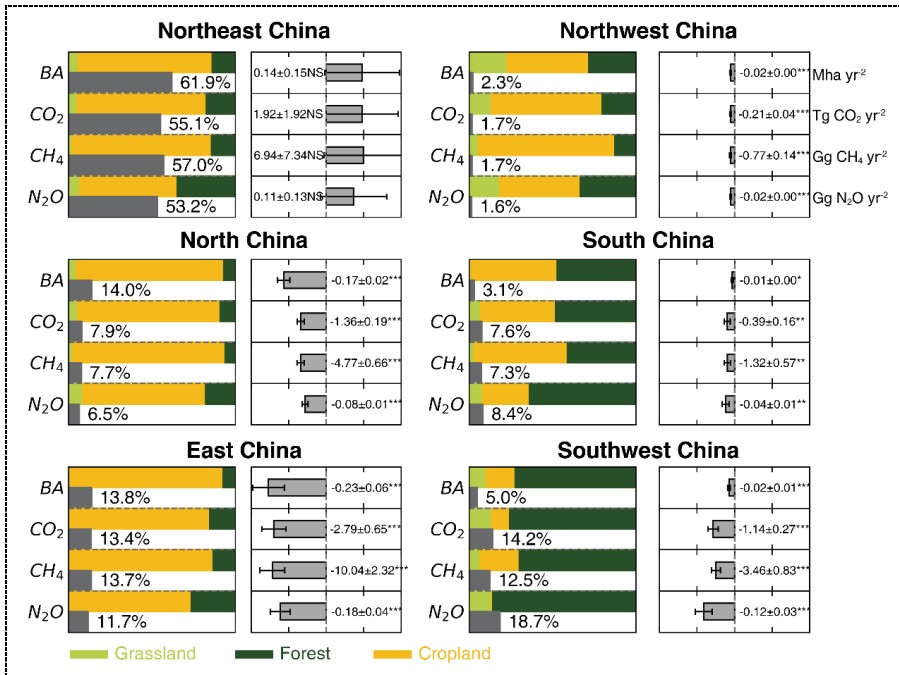

Fig. 4. Regional amounts and trends of wildfire occurrences and emissions, represented in
the relative percentages of national total in colored bars in the left panel and trends in the
dark bars in the right panel. Three colors indicate the different proportions of major land cover
types as aforementioned. Significant trends are denoted by asterisks (*$P$ < 0.1, **$P$ < 0.05,
***$P$<0.01; NS indicates non-significant trends).

East China is another peak region of fire activities both in terms of burned area
and wildfire-induced emissions in our study. This region contains six provinces or
municipalities: Anhui, Jiangsu, Zhejiang, Hunan, Hubei and Shanghai where more than
70% $CO_2$ wildfire emissions came from crop residue burning except for Zhejiang
province. Similar to North China (Hebei, Henan, Shandong, Beijing and Tianjin),
wildfire patterns in East China are featured by high intensity in agricultural-sourced fire
emissions, with a total amount of more than 10 Tg wildfire emitted $CO_2$ and especially
concentrated in the Huanghuai Plain, namely the connection area of Shandong, Henan,
Jiangsu and Anhui. Altogether, these two regions have a half of the national sown area



and grain production and account for 30.8% in cropland burned area, 25.4% in wildfire-
induced $CO_2$ emissions (Fig. 4). During our study period, both of these two regions had
significant declines in agricultural fires at more than -0.22 ± 0.06 Mha $yr^{-2}$ ($P$<0.01) and
-0.17 ± 0.02 Mha $yr^{-2}$ ($P$<0.01) for East and North China, respectively. The decreasing
burned area in cropland led to -2.52 ± 0.64 Tg $CO_2$ $yr^{-2}$ ($P$<0.01), -9.17 ± 2.28 Gg $CH_4$
$yr^{-2}$ ($P$<0.01) and 0.15 ± 0.04 Gg $N_2O$ $yr^{-2}$ ($P$<0.01) in East China. By contrast, there
were an average of 0.59 Mha $yr^{-1}$ in forest fires in the East China, three times higher
than that in North China. This further contributed to significantly more wildfire-induced
emission reduction, reaching 1.57 Tg $CO_2$, 5.03 Gg $CH_4$ and 0.19 Gg $N_2O$ per year.

## 3.3 Comparison with other results

To assess the outcomes of this dataset, we conducted a comparative analysis by
juxtaposing our estimations with those from different studies or products. Our overall
emissions estimates demonstrate moderate values where the amount attributed to
agricultural fires was notably lower compared to former estimates. On average, the
quantities reported in regional to national scale studies were at least three times higher
than our results (Hong et al., 2023; Li et al., 2022; Wu et al., 2018). These studies
employed CYBA as aforementioned that the estimates of burned crop residues is
calculated by the multiplying the crop production derived from statistical data, the grain-
to-straw ratio from field-based analysis, and the proportion of crop residues burned in
the field using empirical summaries. Previous studies had found that the use of very
high residue burning ratios could be the reason for overestimates when compared with
results based on categorized cropland maps (Zhang et al., 2020). Directly utilizing
active fire pixels as proxies for the effects of fire activities can lead to higher values,
thereby contributing to an increase in emission estimates. To address this, we
employed an advanced satellite active fire dataset as a crucial supplementary
observation. This dataset allowed us to refine burned area estimates by reconstructing
external burned regions outside the original burned area data. We achieved this by
using circular kernels centered at active fire records, aligning with the national wildfire
dynamics, which are dominated by agricultural or small-sized fires. Two independent
active fire products and MCD64 burned area product were incorporated as baseline to
make intercomparison (Fig. 5). The sum of pixel area from MOD14 and VIIRS S-NPP
active fire products was translated to 6.77 ± 1.60 Mha and 8.20 ± 2.07 Mha per year
(Giglio et al., 2018, p.6; Schroeder et al., 2014). As a result, the burned area calculation
by directly counting all active fire pixels was at least 27.5% higher than our results.
Expanding to a broader scope, various global fire emission inventories have been
developed using different model settings. We selected four widely used products: (1)
Global Fire Emissions Database (GFED version 4.1s with small fire boosting) (van der
Werf et al., 2017), (2) Fire Inventory from NCAR (FINN version 2.5) (Wiedinmyer et al.,
2023), (3) Global Fire Assimilation System (GFAS version 1.2) (Kaiser et al., 2012) and
(4) Quick Fire Emissions Dataset (QFED version 2.5) (Koster et al., 2015). They
employ either burned area-based approaches (GFED and FINN) or fire energy-based
approaches (QFED and GFAS). Our results maintain similar ranges with other global
products (Fig. 5). The refined calculation for burned area estimates yielded higher



values than the sole use of burned area products and lower values than those only
consisting of active fire products (see details in Methods). Correspondingly, the GHGs
emissions were different as well when active fire-dominated product FINN had higher
estimates than ours. GFED demonstrated 64.3% to 90.3% of the results from
ChinaWED in three GHGs emissions.

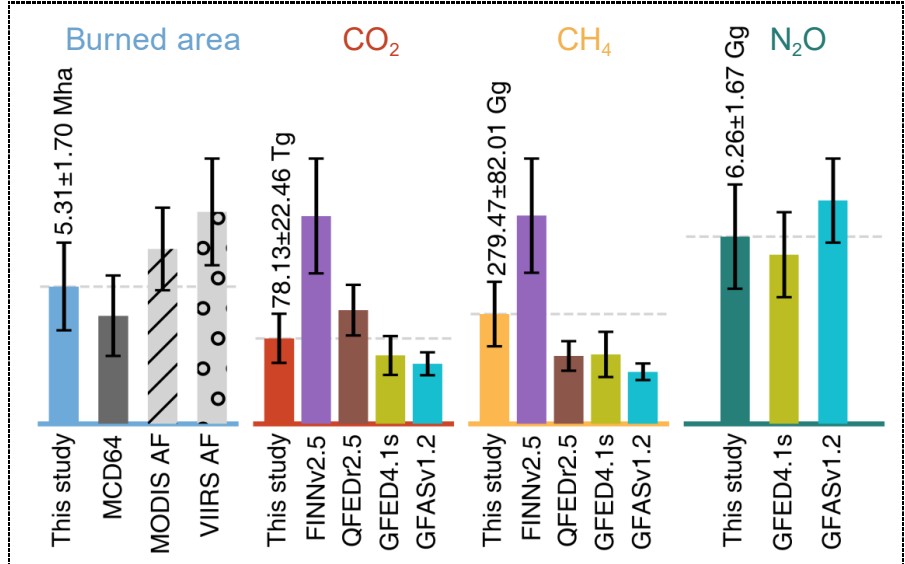

Fig. 5. Comparisons with global fire emission products as well as the burned area baseline calculation with the direct use of satellite datasets. The categories and the products are marked in their titles and x-axis.


## 4. Discussions

### 4.1 Influencing factors of the changes in wildfire seasonal cycles

In China, regulations and policies substantially impact anthropogenic activities
and thus the spatiotemporal distribution of the occurrences of wildfires and emissions.
In agricultural department, the policies have addressed on the issues of straw burning
due to its extensive aerosols and greenhouse gases emissions. In the early 21st
century, a specific law for prevention of air pollution was published, followed by the
releases of regulations on comprehensive utilization of straw (Wu et al., 2018; Zhang
et al., 2015). The national-scale "Air Pollution Prevention and Control Action Plan" was
initiated in 2013, with regional amendment progressively pushing from "legitimate
burning" policy to "strict prohibition" (Geng et al., 2021). Ground-based studies and
satellite-based estimations have documented a rapid decrease in burned area and
emissions across significant regions of the country. Another consequential effect of the
implementation of these banning policies has been the shifts in burning seasons (Ding
et al., 2019; Zhang et al., 2020). Despite Northeast China being the only region with
trends contrary to the national declines, a shift in the primary burning season from
autumn to spring was also observed in this area after 2013 due to the implementation



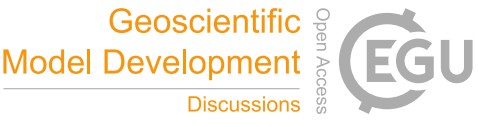

of straw burning bans (Cheng et al., 2022; WANG et al., 2020).
It has been reported that there has been a noticeable decline in the global burned
area driven by the expansion and intensified capital management in agricultural land
use (Andela et al., 2017). Since the beginning of the 21st century, there has also been
a growing emphasis on fire management within both administrative bodies and
scientific communities in China. This evolution has contributed to a more stringent
implementation, particularly in controlling ignition sources in agricultural practices and
forest and grassland areas. From local fire suppression measures to national ignition-
proof initiatives, efforts have been progressively employed to bring forest fires under
control (Chen et al., 2019; Ying et al., 2018). In comparison with forest fire dynamics
reported in previous studies focusing on the first decade of this century, the southern
part of China experienced a significant decline in burned area as well as wildfire-
induced emissions (Wang et al., 2023b; Ying et al., 2018; Zong et al., 2022). Whilst the
establishment and improvement of legal systems and infrastructure for forest and
grassland fire prevention, dealing with uncontrolled transboundary fires remains
challenging. Nationally, an area of 0.07 Mha yr$^{-1}$ was affected within the 10 km buffer
zone near the borders with neighboring countries. This accounted for 1.3% of domestic
burned area and contributed to 1.03 Tg yr$^{-1}$ of $CO_2$, 3.35 Gg yr$^{-1}$ of $CH_4$, and 0.09 Gg
yr$^{-1}$ of $N_2O$.

**4.2 Improvements of ChinaWED to previous studies**
As described in the aforementioned texts, we refined our estimates of emission
factors, fuel loadings and burned area mainly with a set of more localized parameters
and advanced satellite-based observations. Fuel loadings in these previous global
products are mainly derived from biogeochemical models in these global products.
According to the recent studies, the use of abovegound biomass (AGB) as a proxy of
fuel loadings can enable indirect estimations of dry matter and improve fire emission
estimates (Di Giuseppe et al., 2021). We thus used a high-resolution harmonized
carbon density map that was consistently and seamlessly reported across a wide
range of vegetation types based on the relative spatial extent of each type. Emission
factor is a scalar that evaluate the ratio between emission and the total amount of dry
matter that was consumed during burning processes. In this study in addition to the
previously summarized emission factors, we collected the field-based research in
China and neighboring countries and recompiled the values into the new table of
wildfire emission factors for different land cover types. The detailed selection of these
components can be found in Table S2.
Additionally in the estimates of burned area, ChinaWED leveraged the sensitivity
of active fire products with higher spatial resolution and developed a new set of
calculation method that were suitable for smaller fires. The global products had
different frameworks where FINN focuses on active fire detection clusters joined for
the determination of extended burned areas and the burned area from GFED is mainly
derived based on a linear combination of the distribution of active fire and original
burned area data. QFED and GFAS utilize fire energy as the intermediate product to
represent the effects of fires for estimating wildfire-induced emissions. These models




employ empirical continuous functions to incorporate discrete observations and
calculate the temporal integral of fire radiative power (FRP). Furthermore, ChinaWED
is designed for the analysis of wildfire-induced GHG emissions. Most products reported
wildfire-induced $CO_2$ and $CH_4$ emissions while only two of them provided $N_2O$ emission
estimates (Fig. 5).

## Code and data availability

Python code for this model can be obtained from https://zenodo.org/records/13800556
(python version 3.11.6). Key packages used in the code include rasterio (version 1.3.9),
numpy (version 1.25.2), pandas (version 2.1.3) and scipy (version 1.10.1). Fire
products include MCD64A1.061 (doi.org/10.5067/MODIS/MCD64A1.061) and VIIRS
S-NPP active fire (doi.org/10.1016/j.rse.2013.12.008). Aboveground biomass data is
available from doi.org/10.1038/s41597-020-0444-4. Different crop types are available
from double season paddy rice (doi.org/10.3390/rs13224609), single season rice
(doi.org/10.57760/sciencedb.06963), maize (doi:10.6084/m9.figshare.17091653),
winter wheat (doi.org/10.6084/m9.figshare.12003990) and sugarcane
(doi.org/10.3390/rs14051274), respectively.

## 5. Conclusions

Wildfire is one of the most common land-surface disturbances to ecological and
socioeconomical processes. It combusts vegetation and releases greenhouse gases
and aerosols. Employing the burned area-based approach, we featured multisource
fire locations, updated emission factors, and high-resolution fuel load maps to generate
a new China wildfire emission dataset. The wildfire dynamics showed that during the
past decade, an average of 5.31 ± 1.70 Mha burned area, 78.13 ± 22.46 Tg $CO_2$,
279.47 ± 82.01 Gg $CH_4$, and 6.26 ± 1.67 Gg $N_2O$ per year was observed. At the national
scale, the spatiotemporal characteristics of fire occurrences were markedly influenced
by agricultural activities, which contributed to more than four-fifths in area and at least
half in greenhouse gas emissions. The extensive agricultural fires played an important
role in shaping the seasonal cycle of wildfire emissions (Hong et al., 2023; Xu et al.,
2023). Northeast, North, and East China emerged as hotspots for this type of fires,
with the major peak of emissions occurring in mid-spring to early-summer. We
observed rapid and significant decline of burned area and wildfire-induced emissions
in vast areas in China that may be largely attributed to the implementation of fire
prevention and bans on straw burning. Notably, the relative decline rate of burned,
translating to around 5.8% per year, was four times higher than the global average
(Andela et al., 2017). Northeast China was the only region with an opposite trend,
suggesting a situation that requires more adaptive policies rather than mandatory bans.
Compared with estimations by other studies and global products, our results have
moderate values where the mismatches in burned area and estimates of burned crop
residues contributed largely. Overall, the calculation of burned area for small-sized fire
activities and the recalibrated emission factors, tailored for wildfires in China, contribute



to the findings of this study. These results offer new insights into the spatiotemporal
patterns of China's wildfire-induced greenhouse gas emissions and provide important
estimates as a part of the budget for the national terrestrial ecosystems. Future
updates will focus on integrating additional field-based studies and refining the
estimates of various burning processes.

## Author contribution

X.W. designed the study. Z.L. developed the model code and performed the
analyses. Z.L., L.H. and X.W. interpreted the results. Z.L. and H.L. prepared the first
version of the manuscript with contributions from all co-authors.

## Competing interests

The authors declared that none of the authors has any competing interest



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
