# Peer review of "China Wildfire Emission (ChinaWED v1) for the period 2012-2022"

_Geoscientific Model Development, 2024_

## Author Response (AR1)

**Reply to comments**

**Reviewer #1 (Remarks to the Author):**

[**General Comment**] The study introduces the China Wildfire Emission (ChinaWED) model, integrating high-resolution satellite data, updated emission factors, and fuel load maps. It reveals significant seasonal patterns in GHG emissions, with peak emissions months usually from spring to early summer, largely driven by agricultural activities. The findings indicate that the implementation of fire prevention policies has resulted in a substantial reduction in both burned area and GHG emissions over the past decade. The ChinaWED model improves the identification of burned areas and incorporates more detailed parameters for emission factors and fuel load content. Compared to previous global wildfire emission models, it offers reliable estimates of wildfire emissions in China. Therefore, I recommend that this manuscript be considered for publication in Geoscientific Model Development, although some concerns need to be addressed.

RE: We thank the reviewer for taking the time to thoroughly review our manuscript and for providing detailed feedback. Please find below our responses to each of your comments.

[**General Comment**] The authors always emphasize the capabilities of their developed China Wildfire Emission Dataset (ChinaWED) throughout the manuscript. For instance, in the abstract and introduction, they mention statements such as, "This dataset is constructed at monthly and kilometer scale," and "The newly developed product is easily to update with". However, for a journal like GMD, it would be more suitable to frame the manuscript from the perspective of model development and advancements. I strongly recommend that the authors revise the text accordingly. For example, Lines 48-49 could be modified to: "establish a national-scale wildfire emission model to reflect..."

RE: The reviewer's comments have been very helpful in identifying areas for improvement and have significantly enhanced the clarity and quality of the manuscript. We revised our manuscript accordingly not only within the lines mentioned above. Please see the revised manuscript. with revised lines.

**[General Comment]** In the Lines 209-214, the authors state that agricultural wildfire emissions at the national scale have a decreasing trend. However, the description of an increasing trend in agricultural wildfire emissions in Lines 25-26 contradicts the statement that "All vegetation wildfires decreased at different magnitudes" and that "Agricultural fires had been gradually limited and demonstrated a decline in burned area."

RE: We appreciated the comment and have corrected the misdescription. What we originally intended to convey is that the variation in agricultural fires mitigated the overall reduction in the scale of all types of wildfires. Agricultural wildfires and their emissions were also decreasing over this period despite the trends being not significant. Please see the new lines in the revised manuscript.

**[Comment 1]** Method section:

I do not fully understand the method for extracting burned area. Why is it not possible to directly use active fire data to identify the burned area, instead of using them merely as an auxiliary for burned area detection? I suspect there are limitations to this approach, but the authors do not point them out. Additionally, can active fire points be identified as burned areas only if they are located near the burned area? What is the size of the circular buffer used?

RE: We greatly appreciate the reviewer's insightful comments regarding the burned area extraction.

To the first comment: The differences between detecting active fires and burned areas are the primary reason for our use of a combined approach. The active fire detection algorithm incorporates the mid-infrared channel, which, according to Planck's law, is highly sensitive to temperature increase and exhibits significantly higher radiation levels compared to thermal channels. This enables earlier or more sensitive detection, particularly in cases where fires are too small to be captured by the burned area algorithm, or when they burn at relatively low temperatures with incomplete combustion (including smoldering fires, which are common in agricultural fire scenarios). Theoretically, the temporal integration of active fire points during a fire

event should correspond to the burned area at that location. However, as their name suggests, active fire products represent instantaneous observations captured during satellite overpasses and often miss fire detections due to cloud coverage. We therefore combined both active fire and burned area product.

To the second comment: Commission errors, where non-fire hot spots are misclassified as active fire points, are relatively common, especially in areas with highly reflective objects. The spatial proximity of active fire points to burned areas significantly reduces the likelihood of such errors. We employed a reshaped parallelogram, with its center determined by the VIIRS active fire record and its diagonal length corresponding to the resolution of the VIIRS pixel. It also should be noted that the high sensitivity of mid-infrared channel helps the active fire products can detect fires as small as 0.01%-0.1% at pixel level. This approach allows us to account for complex fire conditions while avoiding the overestimation of the actual burned area.

**[Comment 2]** In Lines 168-174, why is the combustion completeness of grassland and cropland related to the percentage of forest cover?

RE: Thanks for the comment. First, we adopted the same method from Wiedinmyer, C.,et al., Geosci. Model Devs., (2011) and Jian Wu., et al., Atmos. Chem. Phys., (2018). Second, the combustion completeness of non-forest pixels are still related to the percentage of forest cover because they are affected by fuel types and combustion conditions. In our study, most pixels comprise mixed vegetation types due to spatial resolution. Consequently, non-forest areas may also exhibit varying degrees of tree coverage. Trees, primarily composed of lignocellulosic fibers, have higher fuel density and moisture content. As a result, combustion in these areas tends to favor smoldering, characterized by lower combustion efficiency. In contrast, areas with low tree coverage, such as grasslands or shrublands, rely on herbaceous fuels as the primary fuel source. These fuels typically have lower density and moisture content, making them more susceptible to flaming combustion (see ref). Additionally, woody fuels tend to burn for longer durations and are more likely to transition into smoldering phases, whereas herbaceous fuels burn for shorter periods and are predominantly associated with flaming combustion.

Ref: Ward, D. E. & Hardy, C. C. Smoke emissions from wildland fires. Environment International 17, 117–134 (1991).

**[Comment 3]** I personally believe that the identification of small-sized wildfires is a highlight of the ChinaWED model. Figures S3 and S4 could be moved into the main text.

RE: Thank you for recognizing our work on identifying small-sized wildfires and for your suggestions regarding the structural changes to the article. We have retained the current structure of the paper and revised the descriptions of how small-sized fires were calculated to enhance clarity and comprehension.

**[Comment 4]** Lines 194-195: the average annual wildfire-induced GHG emissions in China amounted to 78.13 Teragrams (Tg) $CO_2$

RE: Thanks for the comment. We have corrected as suggested.

**[Comment 5]** Line 208: the trend of -0.31 Mha yr-2 differs from the number presented in Figure 1.

RE: Thanks for pointing this out. We have rechecked the data and recalculated the number of the trend.

**[Comment 6]** Lines 212-214: the three types of GHGs in the cropland?

RE: We are sorry for the unclear description of agricultural wildfire emissions. We now have revised the sentence as "…Affected by the variations of agricultural fires, our dataset exhibited a statistically insignificant decline during the study period, with rates of…". Please see the revised manuscript.

**[Comment 7]** Line 295: why are there existing references? Aren't these emissions derived from the ChinaWED dataset?

RE: We are sorry that we placed these references at a misleading location. We intended to incorporate these references as they introduced the wildfire dynamics in Southwest China. Please see the revised manuscript.

**[Comment 8]** Lines 344-346: why does the use of active fire data lead to an

overestimation of wildfire emissions?

RE: We appreciate the reviewer for highlighting this issue. One of the primary causes is that the active fire detection algorithm is highly sensitive to regions with elevated temperatures and high reflectance. Even a small fraction of hot spots, such as flaming vegetation, smoldering fires, or heated high-reflectance objects within a pixel, can trigger active fire detection signals. Consequently, the actual burned area of this kind of places are much smaller than the directly summing these active fire pixels. In other words, the use of active fire products may result in a significantly overestimated burned area compared to other approaches.

**[Comment 9]** Figure 2: the caption "Vertical lines illustrate the peak emissions on different land cover types" is somewhat unclear. The vertical line refers to the months of the year when wildfire emissions peak?

RE: Thank you very much for this comment. We have now revised the caption of this figure.

**[Comment 10]** Figure 4: what does the gray bar in the left panel represent, and what do the numbers represent?

RE: Thank you for pointing this out. The gray bars represent the trends of different items (burned area, GHG emissions). The numbers next to the bars represent their values with their significant levels marked in different forms. We have now revised the descriptions in the new captions of Figure 4 as following:

"… The Y-axis of these subplots represents the four wildfire-related metrics calculated in our study: burned area, $CO_2$, $CH_4$, and $N_2O$ emissions. The colored bars indicate the relative contributions from different land cover types within this region. The dark gray bars represent the proportions relative to the national total, with the corresponding values labeled to the left of the bars. Error bars in the right panel of each subplot depict the trends over the period from 2012 to 2022. …"

**[Comment 11]** The order of the figures in the supplementary material is different from that mentioned in the main text.

RE: We thank the reviewer for pointing this out and we are sorry for the confusion of

orders of figures. We have already rearranged the supplementary figures as well as their index, title and captions as aforementioned suggestions.

**Reviewer #2 (Remarks to the Author):**

[**General Comment**] Lin et al. developed a new dataset for wildfire emissions in China and compared their data with other datasets. I welcome this effort. The approaches used make sense for me. One general question is that I don't see how aboveground biomass data was used a proxy for fuel load. For forests, China's fire is mostly surface fire so what we need is surface fuel load or surface litter. For cropland, I don't know how much Spawn et al. data is reliable in China. I hope that the authors can provide some thoughts in this respect when revising the paper.

RE: We sincerely appreciate the reviewer's insightful comments and constructive suggestions. We have adopted the approach of using above-ground biomass data as a proxy for fuel load, which has been validated in the reference study by Di Giuseppe et al. through comparisons across various models and biomes. This is also one of the key reasons why the relatively high spatial resolution AGB product was used in our current model:

1. The comparisons among GFED, GFAS and the testing results (in the reference paper) indicate that some of the current fire emission products may underestimate total emissions. Burned area-based emission products, such as GFED, incorporate vegetation growth models that simulate the terrestrial carbon cycle. These models account for vegetation characteristics, meteorological data, and fire parameters, and estimate biomass within various carbon "pools." However, they may not be able to accurately and promptly reflect vegetation variability. Meanwhile, FRP-based emission products, such as GFAS, rely on conversion coefficients to estimate fuel loads. Consequently, dry matter estimates derived from AGB are substantially higher, with global increases ranging from 2.7 to 6.1 times depending on the specific AGB algorithm and burned area dataset applied.
2. Through a comparative analysis of simulated Aerosol Optical Depth (AOD) derived from estimated combusted dry matter and independent AOD measurements obtained from the AERONET ground-based network, the reference study demonstrated that dry matter estimates based on AGB outperform those derived from FRP.

Nevertheless, we acknowledge that our current wildfire emission models for China still have limitations, particularly in estimating fuel loads for dominant surface fire types and in the selection of AGB input datasets. The average AGB in croplands, as derived from Spawn et al.'s data, is 7.07 ($\pm$1.35) Mg/ha (see figure 1 below). These results were calculated based on global crop yield data with significantly coarser spatial resolution, which may not adequately represent the diverse agricultural

patterns in fragmented, smallholder-dominated croplands. Furthermore, given that the high-spatial-resolution AGB dataset was only available as a static input, our emission estimates were still predominantly influenced by wildfire variability. In future updates, we plan to explore the feasibility of incorporating dynamic aboveground biomass products to enhance the accuracy of emission estimates.

[Figure]

Figure 1 Aboveground biomass within cropland area.

Ref: Di Giuseppe, F., Benedetti, A., Coughlan, R., Vitolo, C. & Vuckovic, M. A Global Bottom-Up Approach to Estimate Fuel Consumed by Fires Using Above Ground Biomass Observations. Geophys. Res. Lett. 48, e2021GL095452 (2021).

**[Comment 1]** Line 106-108: justifications are needed on why these two products are selected out of other potential products.

RE: Thanks for this comment. Currently global active fire products include MOD14/MYD14 series and VNP14 series. We selected the product generated from VIIRS due to its higher spatial resolution, which is more effective in agricultural fires. There are more global burned area products including MCD64 from NASA and Fire_CCI from ESA. We selected MCD64 because of the following two reasons: 1) MCD64 is generated continuously at monthly scale and covers the whole study period while Fire_CCI is provided only from 2001-2020. 2) In previous study areas in southwest China (see ref), the performance of MCD64 is slightly higher than that of Fire_CCI. Hence, the combination of these two products can provide enhanced observation at higher resolution compared with previous studies.

Ref: Fornacca, D., Ren, G. & Xiao, W. Performance of Three MODIS fire products (MCD45A1, MCD64A1, MCD14ML), and ESA Fire_CCI in a mountainous area of

Northwest Yunnan, China, characterized by frequent small fires. Remote Sensing 9, 1131, (2017).

**[Comment 2]** Line 117: I don't see how Fig. S1 shows the system⋯ Do you mean Fig. S3? But Fig. S3 is not very clear either.

RE: We sincerely appreciate your valuable comments pointing out the unclear expression regarding the processing of these two products. We have carefully revised the manuscript and updated the corresponding figure descriptions to enhance clarity and understanding. Additionally, the rationale for using both active fire and burned area products has been addressed in our response to Reviewer #1's comments.

**[Comment 3]** Line 125: Fig. S3 and S4 need more explanations. The current ones are not easy to follow.

RE: Thank you for this comment. We have now revised the descriptions in manuscript and fig S3 (now fig S1) as suggested in comment#2. The content of description of fig. S4 (now fig S2) is also revised for better understanding of how the industrial hot spots are removed.

**[Comment 4]** Line 131: "burned areas" => you could name them as false active fire detections.

RE: Thanks for the comment. We have corrected as suggested.

**[Comment 5]** Line 132-133: the resampling to 1km is critical as the authors argued that cropland residual burning is very small in China. How this resampling is done and how will it affect the burned area product?

RE: We appreciate the reviewer's comment. The resampling procedure was performed following the sampling instructions outlined in Table S1. To calculate the burned area of an active fire pixel, we employed a reshaped parallelogram, with its center determined by the VIIRS active fire record and its diagonal length corresponding to the resolution of the VIIRS pixel. In addition, we estimated the burned fraction of a single pixel with the conceptual model that incorporates both burned area and active fire results with their overlapping areas filter out. Given the

inherent differences between the VIIRS and MODIS instruments, we aligned these products to a 1km resolution to minimize potential biases. It is important to note that while the incorporation of VIIRS has enhanced the detection of small agricultural fires, directly summing these detections inevitably introduces overcounting of burned areas. To evaluate this, we tested a resampling approach by aggregating VIIRS active fire records as a proxy for burned area estimation. This approach resulted in an annual average overestimation of 32% (121%~153% to synchronous results from our calculation), potentially further inflating the burned area estimates.

**[Comment 6]** Section 2.3 (line 156-157): I don't see the need why AGB and land cover products have both 300m resolution while you decide to finally work on 1km resolution, followed by resample burned area to 1km?

RE: Thank you for your comments. We chose 1km as the final output resolution primarily to unify the spatial resolution across different datasets while balancing computational efficiency and analytical accuracy. Our input datasets include 300m land cover products and AGB, 375m VIIRS active fire records, and 500m MODIS burned area products. Using the original resolutions of these datasets would result in inconsistent spatial resolutions, complicating the data processing. For large-scale regional analysis, our 1km-resolution-results provide a good trade-off between preserving analytical accuracy and capturing the relevant spatial patterns while avoiding potential noise and errors associated with resolution mismatches. In future studies, we plan to incorporate higher-resolution fire-related input data to further improve performance.

**[Comment 7]** Table S1: the setting of CC seems a little arbitrary. Justifications are needed.

RE: Thanks for the suggestion and we admit that the settings of combustion completeness are still far from accurate simulating the burning process. The methodology of piecewise function rather than fixed threshold was adopted from Wiedinmyer, C.,et al., Geosci. Model Devs., (2011) and Jian Wu., et al., Atmos. Chem. Phys., (2018).

**[Comment 8]** Fig. 5: it does not quite make sense to use MCD64 for BA comparison because it is already used in the authors' dataset? The other two are based on active fire and hence not a good source either. Why not using GFED5? ESA CCI

burned area datasets?

RE: Thank you for your comment. We generated comparison plots to demonstrate that our method captures more fire information than using a single burned area product alone, while effectively avoiding commission errors in high-reflectance pixels, particularly in rural areas of China (see ref). Therefore, we used the original input dataset for comparison. Additionally, MODIS active fire records were employed as supplementary evidence to highlight potential errors that may arise from this type of dataset. In this study, we did not compare the burned area estimation with GFED5 as they represent different types of datasets. Our ChinaWED calculated the burned area directly from satellite observations at medium spatial resolution. GFED5, on the other hand, should be considered as a hybrid product combining a series of satellite products with an order of magnitude higher resolution (e.g., Landsat and Sentinel), which potentially includes more fire information. Regarding the ESA CCI product, the difference in temporal coverage prevents direct comparisons, as discussed in our previous comments.

Ref: Zhang, T., Wooster, M. J. & Xu, W. Approaches for synergistically exploiting VIIRS I- and M-Band data in regional active fire detection and FRP assessment: A demonstration with respect to agricultural residue burning in Eastern China. Remote Sensing of Environment 198, 407–424 (2017).

**[Comment 9]** Fig. 1: I am surprised to such a huge interannual variation in BA (about five times!) while it is still dominated by agricultural fires which are human dominated. Did the authors think about the reason?

RE: We appreciate the comments regarding the interannual variation of burned areas. We have now added discussion on the huge interannual variation of BA and its relevant wildfire emission.

The observed variation in fire activity is largely influenced by the implementation of policies banning straw burning and continuously enhanced regulations aimed at controlling forest fires. Along with the initiated "Air Pollution Prevention and Control Action Plan" in 2013, the second revision of the Atmospheric Pollution Prevention and Control Law in 2015 introduced additional provisions prohibiting straw burning, which came into effect at the beginning of 2016. This period saw a substantial reduction in the total burned area (before 2016: 6.46Mha/yr ==> after 2016: 3.89Mha/yr).

From a regional perspective, the burned area sizes have significantly decreased

across all regions except for Northeast China (before 2016: 3.87 Mha/yr ==> after 2016: 1.13 Mha/yr). The majority of this reduction can be attributed to the burning of agricultural residues, which accounted for over 90% of the decline during 2012–2022. Specifically, the burned area within croplands decreased at a rate of -0.46 Mha/yr$^2$, while the total burned area declined at -0.51 Mha/yr$^2$. By categorizing the sources of the estimated burned area, we found that most of the reductions in cropland burned area were linked to declines in raw burned area data products rather than the scattered active fire points (Figure 2). This suggests that the enforcement of regulations and legal measures has played a crucial role in curbing agricultural burning on a considerable scale. Northeast China exhibited a different trend during the study period, with agricultural fires in Liaoning and Jilin showing contrasting patterns (Figure 3). Burned areas increased significantly from 0.43 Mha/yr before 2016 to 1.05 Mha/yr after 2016, making these regions one of the primary contributors to the observed large variations at national scale.

[Figure]

Figure 2 Burned area in all regions except for Northeast China. The sum of these regional burned area plotted in green and gray lines, representing the agricultural fires and burned area across all vegetation types. We separated the counts from raw burned area and active fire products in solid and dashed lines. Rapid declines can be found over 2012-2016 with the majority of them located in cropland.

[Figure]

Figure 2 Burned area in Northeast China. Regional agricultural fires and all vegetation types in green and gray lines. We added the variations of agricultural fires in Liaoning and Jilin in dashed/dotted lines.